# The Association of Cerebral Blood Flow Measured Using Extracranial Carotid Ultrasound with Functional Outcomes in Patients with Anterior Circulation Large Vessel Occlusion After Endovascular Thrombectomy—A Retrospective Study

**DOI:** 10.3390/neurolint17050067

**Published:** 2025-04-25

**Authors:** Xin-Hong Lin, Kuan-Wen Chen, Chung-Fu Hsu, Ting-Wei Chang, Chao-Yu Shen, Hsin-Yi Chi

**Affiliations:** 1Department of Neurology, Chung Shan Medical University Hospital, Taichung 40201, Taiwan; u101022009@gmail.com (X.-H.L.);; 2Department of Neurology, Chung Shan Medical University, Taichung 40201, Taiwan; 3Department of Medical Imaging, Chung Shan Medical University Hospital, Taichung 40201, Taiwan; shenchaoyu@gmail.com

**Keywords:** endovascular mechanical thrombectomy, anterior circulation large vessel occlusion, carotid sonography index, cerebral blood flow, post-EVT outcomes

## Abstract

**Background**: Endovascular mechanical thrombectomy (EVT) is regarded as the standard treatment for acute ischemic stroke with large vessel occlusion. Few studies have examined the evolution of cerebral flow after the acute stage of ischemic stroke. In this study, we examined the association of functional outcomes with cerebral blood flow by extracranial carotid sonography during the subacute phase after EVT and multiple prognostic variables. **Methods**: We conducted a single-center, retrospective, observational study between January 2018 and June 2023. Patients with acute stroke resulting from anterior circulation large vessel occlusion who underwent EVT were included. All patients underwent carotid sonography in the second week after EVT. Patients with fair (modified Rankin Scale [mRS]: 0–3) and poor outcomes (mRS: 4–6) were compared to determine the association between and identify the predictors of these factors and functional outcomes. **Results**: A total of 89 patients were included (female: 38 (42.7%); mean age: 69.45 ± 13.59 years). Multivariable logistic regression analysis revealed that three factors were independent predictors of fair outcomes: (1) the Alberta Stroke Program Early CT Score (odds ratio [OR]: 1.79; 95% confidence interval [CI]: 1.16–2.78; *p* = 0.009); (2) Thrombolysis in Cerebral Infarction 2b to 3 (OR: 4.91; 95%CI: 1.10–21.89; *p* = 0.037); (3) the ratio of treatment-side blood flow between the internal carotid artery and common carotid artery (QTI/QTC, OR: 45.35; 95% CI: 1.11–1847.51; *p* = 0.04). **Conclusions**: The ratio of QTI/QTC is a clinically relevant parameter as a potential predictor of favorable outcomes. This parameter can be used to formulate patient prognostic scores and help clinicians determine whether adequate cerebral perfusion is maintained during the subacute phase.

## 1. Introduction

Endovascular mechanical thrombectomy (EVT) is the standard treatment for acute ischemic stroke caused by large vessel occlusion (LVO) [1,2,3]. Predictive factors for favorable outcomes after EVT have been extensively studied. Key factors include young age, lower initial National Institutes of Health Stroke Scale (NIHSS) scores, fewer early ischemic changes on non-contrast computed tomography (CT), a smaller infarction core, good collateral status, shorter procedure duration, and successful recanalization [3,4,5,6].

Even when EVT is used for treatment, stroke continues to evolve after the acute phase. Various post-procedural factors substantially influence the course of this cerebral vascular event, such as cerebral autoregulation, the development of intracranial hemorrhage, the role of collateral circulation, surgical interventions for ischemic edema, target artery re-thrombosis, and underlying metabolic disorders [7,8]. Certain post-procedural factors, such as postoperative 24 h NIHSS scores, hemorrhagic conversion, decompressive hemicraniectomy, and the length of a hospital stay, may serve as predictive variables for outcomes [5,9,10]. However, a post-EVT prediction tool aimed at guiding prognostic expectations has not been well established [5,9,11].

Traditionally, brain imaging modalities are used 24 to 48 h after EVT to monitor infarction size and detect complications [1,2]. However, these imaging modalities do not fully capture dynamic changes in cerebral blood flow, particularly in the interaction between the ischemic regions, the penumbra, and healthy cerebral arteries [12,13]. Deficiencies in predictors of functional outcomes after the acute phase of EVT lead to challenges for physicians in accurately estimating clinical prognosis and organizing the next treatment plans.

Cerebral sonography facilitates the real-time assessment of cerebral blood flow (CBF), providing valuable insights into hemodynamic changes in target and collateral arteries following EVT. CBF correlates with cerebrovascular reserve, and high CBF in the target artery is associated with favorable outcomes in patients who undergo mechanical thrombectomy [12,14]. In anterior circulation (AC), the net flow volume (FV) of the middle cerebral artery (MCA) and the anterior cerebral artery (ACA) is equal to the FV of the internal carotid artery (ICA). A high FV in the treatment-side ICA represents greater CBF. In this study, we focused on the FV of the target AC after recanalization. Following this rationale, we measured CBF using sonography in the second week after EVT and compared the flow ratios of the treatment-side ICA to the treatment-side common carotid artery (CCA) and to the non-treatment-side ICA. In this study, we aimed to evaluate the correlation between the CBF ratio and clinical outcomes in patients with anterior circulation–LVO (AC-LVO) post-EVT.

## 2. Methods

This retrospective observational study was approved by the Institutional Review Board of Chung Shan Medical University Hospital, Taichung, Republic of China. The requirement for informed consent was waived because of this study’s retrospective nature.

### 2.1. Study Population

We reviewed the data of 1789 consecutive in patients with acute ischemic stroke who were admitted to Chung Shan Medical University Hospital between January 2018 and June 2023. We only included patients who had undergone EVT with a stent retriever or a combined approach for AC-LVO. The EVT criteria in this study were an acute ischemic stroke due to a large vessel occlusion, with an Alberta Stroke Program Early Computed Tomography Score (ASPECTS) ≥6, a National Institutes of Health Stroke Scale (NIHSS) score ≥6, and a stroke onset time of less than 6 h [1]. We defined AC-LVO as an occlusion in the ICA, an M1 segment of the MCA, or a proximal M2 segment of the MCA and ACA.

All patients had undergone National Institutes of Health Stroke Score (NIHSS) evaluation [15], modified Rankin Scale (mRS) assessments, ASPECTS [16], and CT angiography (CTA) assessments before EVT. We collected medical and laboratory data, including age, sex, hypertension, diabetes mellitus status, renal function, lipid profiles, anticoagulant use, malignancy, pre-stroke functional status, and histories of previous stroke or transient ischemic attack. A combination of intravenous thrombolysis with recombinant tissue-type plasminogen activator (r-tPA) administration or not before EVT was recorded. All patients underwent carotid sonography in the second week after EVT, and their mRS scores were evaluated 3 months after the procedure. Patients who did not complete their sonographic examination or were lost to follow-up were excluded from this study.

### 2.2. Data and Examination

ASPECTS values were derived from 5 mm thick non-contrast CT and multi-phase CTA, independently evaluated by 2 neuroradiologists.

Recanalization success was evaluated using the Thrombolysis in Cerebral Infarction (TICI) grading system on the final control angiogram. In this study, revised TICI scores [17] were applied, including 0 (no flow), 1 (penetration but not perfusion), 2A (some perfusion with the distal branch filling of <50% territory visualized), 2B (substantial perfusion with the distal branch filling of ≥50% of territory visualized), 2C (near-complete perfusion except for slow flow in a few distal cortical vessels or the presence of small distal cortical emboli), and 3 (normal flow).

The color-coded carotid duplex of extracranial vessels was performed by experienced technicians using an EPIQ Elite system (Philips Health care, Andover, MA, USA). Routine measurements included thorough examinations of the bilateral CCAs, ICAs, and extracranial carotid arteries (ECAs). The angle between the ultrasound beam and the direction of blood flow was manually adjusted. Ultrasonographic data, including the diameter, FV, mean velocity (MV), and pulsatility index (PI) of the CCAs, ECAs, and ICAs, were recorded and analyzed.

### 2.3. Variable Abstraction and Management

To analyze CBF in the treatment-side arteries, we calculated the ratios of blood flow between the treatment-side ICA and treatment-side CCA (abbreviated as the QTI/QTC ratio) and between the treatment-side ICA and non-treatment-side ICA (abbreviated as the QTI/non-QTI ratio). Subsequently, we evaluated clinical outcomes using the mRS 3 months after stroke. Using our predictive model, we classified outcomes as either fair (mRS score: 0–3) or poor (mRS score: 4–6).

To validate our sonography index, we utilized well-established predictors for favorable functional outcomes [3,5,18,19], including age < 70 years, ASPECTS value > 8, NIHSS score < 15, and TICI 2b to 3. These variables were used to confirm the significant factors of fair and poor outcomes in this study.

### 2.4. Statistical Analysis

Continuous data were expressed as the means and standard deviations (SDs) or medians and interquartile ranges (IQRs). Means and medians were compared using the *t*-test and Mann–Whitney U-test, respectively. Group differences in demographic data were evaluated using the Chi-square (*x*^2−^) test, while Spearman’s correlation analysis was conducted to identify correlations between prognostic factors and clinical outcomes. Binary logistic regression analysis was used to examine the factors associated with fair outcomes, with the results summarized as odds ratios (ORs) and 95% confidence intervals (CIs). Clinically relevant variables and key demographic indicators with a *p*-value less than 0.2 in the univariate analysis were included in the multivariate logistic regression analysis. Receiver operating characteristic (ROC) curve analysis was conducted to determine the cut-off values for the carotid artery sonographic data. All statistical analyses were conducted using SPSS version 17 (SPSS, Chicago, IL, USA). Two-sided *p*-values of less than 0.05 were considered statistically significant.

## 3. Results

A total of 1789 patients were admitted for ischemic stroke between January 2018 and June 2023. Among these patients, 172 underwent EVT, with 118 meeting the criteria for AC-LVO. Ultimately, 89 patients who completed their follow-up were enrolled in this study (Figure 1). We excluded 29 patients who did not complete their sonography exam or follow-up. The comparison of baseline characteristics between these 29 patients and our enrolled cases can be found in Table 1.

### 3.1. Comparison Between Enrolled and Excluded Patients

The baseline characteristics of the enrolled and excluded patients are shown in Table 1.

A higher proportion of NOACs administration, a higher ASPECTS, a higher portion of favorable recanalization (TICI grades 2b-3), and a lower initial NIHSS score in enrolled patients were observed compared to excluded patients.

### 3.2. Baseline Variables and Correlation Analysis

Table 2 presents the baseline characteristics of the patients, including demographic characteristics, risk factors, medication history, initial NIHSS scores, ASPECTS, mRS scores (at baseline and at 3 months), TICI scores, and sonographic findings.

Correlation analysis revealed a significant association between clinical outcomes (mRS score at 3 months) and several variables, including age, r-tPA administration, initial NIHSS score, ASPECTS, a TICI of 2b to 3, and QTI/non-QTI and QTI/QTC ratios (Table 2).

### 3.3. Predictors of Fair Outcomes

Patient outcomes were classified as either fair (mRS: 0–3) or poor (mRS: 4–6) 3 months after EVT. A total of 53 patients (59.5%) achieved fair outcomes, who exhibited higher ASPECTS values, higher rates of successful recanalization (TICI of 2b to 3), higher rates of r-tPA administration, and high QTI/QTC ratios (Table 3).

Univariate analysis revealed that r-tPA administration, the ASPECTS, a TICI 2b to 3, and the QTI/QTC ratio were significantly associated with fair outcomes. After adjusting the potential associated factors, including age, sex, cancer, initial NIHSS score, r-TPA administration, ASPECTS, and a TICI of 2b to3 in the univariate analysis (*p* < 0.20), in the multivariate analysis, the ASPECTS (OR: 1.79; 95% CI: 1.16–2.78; *p* = 0.009), a TICI of 2c to 3 (OR: 4.91; 95% CI: 1.10–21.89; *p* = 0.037), and the QTI/QTC ratio (OR: 45.35; 95% CI: 1.11–1847.51; *p* = 0.04) remained independent predictors of fair outcomes (Table 3). Model analysis revealed fair calibration (*p* = 0.890 in the Hosmer–Lemeshow test) and discrimination (C statistics = 0.834); meanwhile, the ROC curve analysis (Figure 2) of the QTI/QTC ratios revealed an optimal threshold of 0.60, with an area under the curve (AUC) of 0.641 (95% CI: 0.52–0.76; *p*: 0.024).

### 3.4. Validation of Outcome Predictors, Including Sonography Index

To establish a validation model, a QTI/QTC ratio >0.6 and other independent predictors stated in the Methods Section were used for multivariate logistic regression analysis.

Among our patients, the administration of r-TPA (OR 4.25; 95% CI 1.25–14.46; *p* = 0.002) and a QTI/QTC ratio >0.6 (OR 3.83; 95% CI 1.28–11.48; *p* = 0.016) served as significant predictors of fair outcomes (mRS score, 0–3) at 90 days (Table 4). This model demonstrated fair calibration (*p* = 0.861 by the Hosmer–Lemeshow test) and discrimination (C statistic = 0.812).

## 4. Discussion

Since 2015, EVT has become the standard of care for patients with acute ischemic stroke caused by large vessel occlusion [1,2,20,21].

In spite of advancements in reperfusion therapies and the implementation of established pre-procedural selection criteria for EVT, a large portion of acute ischemic stroke patients still experience severe morbidity or mortality [4,5]. A deficiency in predictors of functional outcomes after the acute phase of EVT poses challenges for physicians in accurately estimating clinical prognosis and planning subsequent treatments.

To address these challenges, we aimed to identify clinically useful factors and predictors correlated with functional outcomes after the acute period of stroke.

Our preliminary analysis revealed that age, r-tPA administration, initial NIHSS score, ASPECTS, a TICI of 2b to 3, and QTI/non-QTI and QTI/QTC ratios measured using extracranial carotid ultrasonography (Table 2) were significant factors associated with outcomes at 3 months. Unlike the medical comorbidities highlighted by the THRIVE score [22], the risk factors for ischemic stroke, such as type II diabetes mellitus, dyslipidemia, hypertension, smoking, and atrial fibrillation, did not correlate with the functional outcomes in this study.

Several predictive scoring systems have been developed to estimate outcomes following EVT, including the Pittsburgh Response to Endovascular Therapy score (PRE) [23], Computed Tomography for Late Endovascular Reperfusion Thrombectomy Score (CLEAR) [24], Bronx Endovascular Thrombectomy Score (BET) [25], baseline disability, age, NIHSS, delay from last known normal time (BAND) [26], and pre-stroke disability, age, NIHSS, delay from last known normal time, ASPECTS (PANDA) [27] scores. Each score offers unique advantages and clinical relevance. For example, the PRE score focuses on age, NIHSS, and ASPECTS, but it does not incorporate traditional clinical risk factors. The CLEAR score was developed specifically for late-window EVT and, therefore, does not align with our inclusion criteria. The BET score emphasizes peri-procedural details such as anesthesia type, puncture-to-perfusion time, and post-EVT NIHSS—variables that were unavailable due to limitations in our recorded data. Both the BAND and PANDA scores require the precise documentation of the delay from the last known normal time, a variable that was inconsistently recorded in our registry. Given these limitations, we selected the THRIVE-EVT score for comparison in our study. This score incorporates age, NIHSS, and common vascular comorbidities (hypertension, diabetes mellitus, and atrial fibrillation) available at the initial presentation, making it highly applicable to our retrospective dataset.

The analysis of predictive factors in patients with fair outcomes (mRS: 0–3) at 3 months confirmed the importance of the ASPECTS, recanalization success, and QTI/QTC ratio (Table 3). In the validation model (Table 4), most of the predictors were clinically relevant (*p* < 0.1) but did not meet the statistical significance criteria of the study outcomes. Both r-tPA administration and a QTI/QTC ratio >0.6 served as key predictors.

In a series of studies, the clinical application of sonography following EVT in AC-LVO, particularly transcranial Doppler (TCD) and transcranial color-coded sonography (TCCS) [28,29,30,31], was discussed, in which sonographic parameters were considered useful indicators in the early detection of complications in post-EVT settings and predictors for clinical outcomes [29,31]. The timing of sonographic assessments in these studies typically fell from within the first 24 h to one week post-EVT. The peak systolic velocity (PSV) in the MCA was the main checking parameter, and its elevation in the acute stroke stage (within 48 h) was associated with the risk of hyperperfusion or hemorrhagic transformation [28,29,30]. Additionally, if the MCA PSV returned to normal 1 week after EVT in hemorrhagic patients, they tended to have better outcomes [31]. In some studies, it was suggested that TCD-guided blood pressure control in the acute stage post-EVT can reduce the incidence of early neurological deterioration and improve outcomes, especially in patients with abnormal TCD parameters [30]. Different from TCD assessments, in our study, the timing of sonographic assessment was in the second week, when the MCA PSC gradually returned to a normal range [30,31], and the checking parameter was the extracranial FV.

The QTI/QTC and QTI/non-QTI ratios are novel indices indicating the status of cerebral perfusion and are significantly associated with clinical outcomes. In anterior circulation, the ICA-FV is calculated as the sum of the MCA and ACA FVs, while the sum of the ICA and ECA FVs is equal to the CCA FV. Successful and persistent recanalization post-EVT in a patient with AC-LVO would result in more increases in flow in the ipsilateral-side ICA. Therefore, a higher ratio of QTI/QTC is expected. This ratio can be used to reflect the status of subacute perfusion and the effectiveness of target arterial circulation after EVT [12]. In this study, we discovered that the QTI/QTC ratio served as an independent predictor of fair outcomes after 90 days (mRS score: 0–3, Table 3 and Table 4).

After vessel recanalization, three conditions may develop [10,32,33]. The first is persistent stenosis or occlusion, which indicates hypoperfusion. The second is the attainment of autoregulation within the brain, which indicates the restoration of adequate circulation. The third is hyperperfusion syndrome, which indicates tissue injury due to a sudden increase in cerebral blood flow. The final clinical outcome depends on how the targeted cerebral flow evolves among the three conditions. In addition to vessel recanalization after EVT, blood pressure fluctuation [30,34], collateral flow supply [12,14], ischemic oxidative stress [35], cell damage, and inflammatory reactions [36] are essential factors contributing to the pathophysiology and outcomes of stroke [12,14,35,36,37,38]. Although direct examinations of complex interactions are challenging in clinical studies, the timeline of blood pressure changes after acute stroke may provide valuable insights [34,38,39,40]. According to the literature, reactive blood pressure typically substantially decreases after the first week of cerebral infarction, with minor changes thereafter [30,39,40]. In patients whose blood pressure remains high after the first week, severe stroke with poor outcomes may occur [31,38]. In clinical practice, initial reperfusion outcomes can be rapidly evaluated through CT or magnetic resonance imaging within 48 h after treatment [12,14,37]. In this study, given the correlation between blood pressure and stroke outcomes, we used the cerebral blood flow volume obtained through sonography in the second week after stroke, a stage that represents the gradual normalization of cerebral autoregulation after the recanalization of LVO [40], to evaluate the total effect of recanalization, cerebral autoregulation, and perfusion on surviving tissue. We regarded the flow ratio between the ICA and CCA as the flow required from the treated territory in the ipsilateral cerebral artery. The FV obtained in the second week after stroke is considered valuable because it may reflect that the neurons supplied by the targeted artery survived after the ischemia and inflammatory storms in the acute phase of stroke [13]. In healthy individuals, the FV ratio between the ICA and the CCA is between 0.58 and 0.65 [41,42]. In this study, the absolute value of the QTI/QTC ratio in patients with mRS scores of 0–3 and 4–6 was 0.66 and 0.57, respectively, and the ROC curve analysis revealed an optimal threshold of 0.6. Although the ratio was very close between these two groups, the greater ICA FV in the treatment side indicated more cerebrovascular reserve and high CBF in the target artery. We showed that the ratio of QTI/QTC is associated with patient outcomes (Table 2) and can be used as a predictive factor for EVT outcomes (Table 3 and Table 4).

According to our rationale, the role of collateral flow from the ipsilateral ECA or the circle of Willis in supporting perfusion is ignored. Collateral circulation is crucial in infarct evolution and evaluating eligibility for endovascular treatment. However, the link between collateral flow and clinical outcomes in this therapy remains debated [6,12,14,43]. Kim et al. [14] found that collateral grades and reperfusion status were strong predictors of clinical outcomes in acute MCA stroke with a larger artery occlusion. Lee et al. [44] conducted a systematic review emphasizing the significance of collateral circulation in patients undergoing EVT for LVO. Their findings suggested that robust pretreatment collaterals are associated with reduced infarct growth, higher recanalization success, and a decreased risk of hemorrhagic transformation. Hassler et al. [45], who studied LVO stroke patients with underlying carotid artery stenosis (CAS), found that pre-existing CAS was an independent predictor for favorable collateral status and more frequent adverse events after EVT; however, post-interventional infarct size and functional 90-day outcomes were not different between CAS and non-CAS patients. Villringer et al. [12], who collected data on collateral grades within 24 h after stroke onset and follow-up measurements 24 h later, stated that recanalization, but not collateral flow grades, was independently associated with good clinical outcomes. Fluctuations in collateral flow over time, coupled with the steal-like effect by the recruitment of collateral in adjacent regions, explained unstable blood supply and unreliable support for hypoperfusion tissue [43]. Extending the findings mentioned above, the TI/TC ratio, focused on the flow in the territory of the target artery, represented an appropriate parameter for the recanalization of perfusion and significantly correlated with the clinical outcomes in our study.

This study had several limitations. First, we used a retrospective design, which is inherently associated with a risk of selection bias. Initially, 118 patients met the criteria for AC-LVO. We excluded 29 patients who did not complete their sonography exam or follow-up. This exclusion possibly introduced selection bias since these patients exhibited traits linked to poor outcomes, such as low ASPECTS, low recanalization rates, and high initial NIHSS scores. Second, this study did not include other potential prognostic variables like blood pressure, sugar levels, and incomplete data. It was also conducted at a single center with a small cohort, which may limit the generalizability of its findings. Third, sonography was performed from day 7 to day 14 after EVT, potentially introducing variability in hemodynamic evolution. Fourth, we did not exclude patients with hemorrhagic conversion, which resulted in different flow components in our study. Lastly, the inclusion of multiple variables in the regression analysis, combined with the small sample size, increased the risk of overfitting; the wide confidence interval for the QTI/QTC ratio in the logistic regression may have reflected this deficiency. Despite these limitations, our findings lay the foundation for the evaluation of post-EVT recovery, providing clinicians with an additional metric to identify patients who are most likely to benefit from EVT. Future studies should validate our findings with larger multicenter cohorts and provide more direct and resourceful guidance for researchers in the field.

## 5. Conclusions

In this study, we identified several relevant factors for patient outcomes after EVT, including age, r-tPA administration, initial mRS score, ASPECTS, recanalization success, and QTI/QTC ratio. Of these variables, the QTI/QTC ratio particularly holds promise as a clinically relevant factor for patient outcomes and an independent predictor of fair outcomes in patients with ischemic stroke. To our knowledge, this is the first study to find a correlation between clinical outcomes and cerebral blood flow measured using ultrasound in patients post-EVT. An important practical implication is to refine pre- and post-EVT prognostic scores and guide the treatment plans of clinicians.

## Figures and Tables

**Figure 1 neurolint-17-00067-f001:**
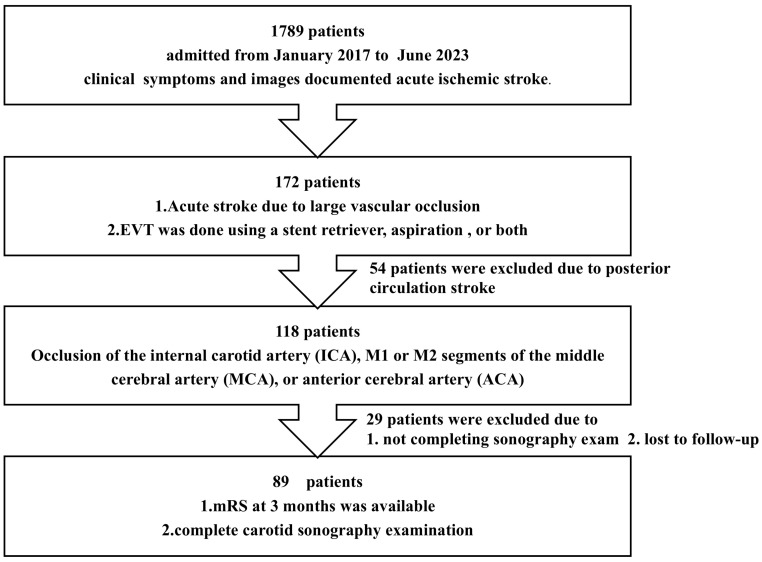
A flowchart illustrating the workflow for the inclusion criteria of this study. EVT, endovascular mechanical thrombectomy; mRS, modified Rankin Scale.

**Figure 2 neurolint-17-00067-f002:**
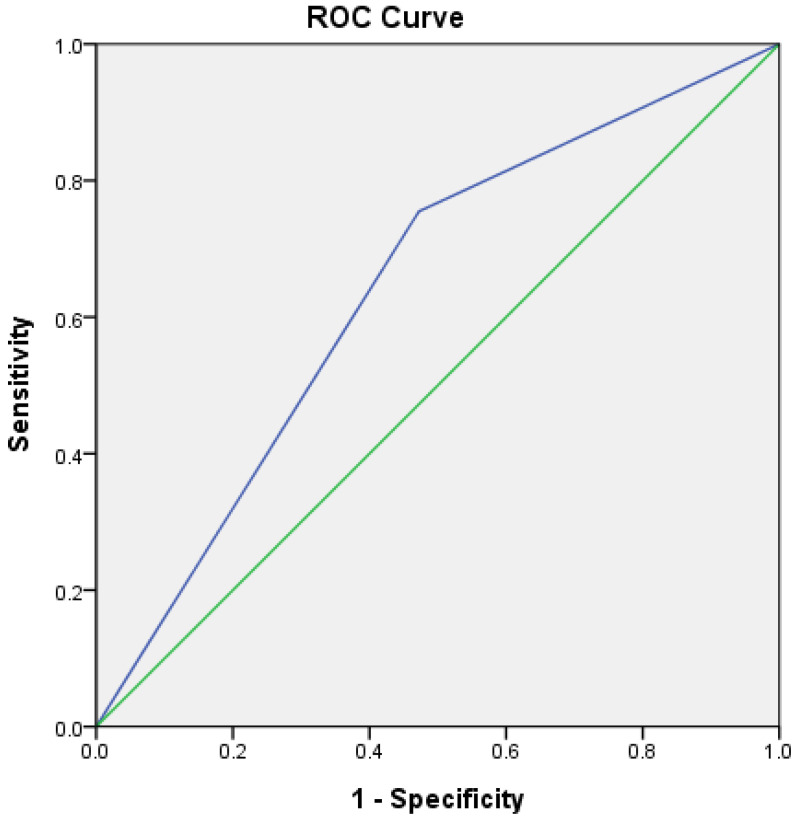
Area under the receiver operating characteristic curve of QTI/QTC ratio 0.6 with an area under the curve (AUC) of 0.641 (95% CI: 0.52–0.76; *p*: 0.024). The blue line represents the QTI/QTC ratio, and the green line represents the reference line.

**Table 1 neurolint-17-00067-t001:** The baseline characteristics of the enrolled and excluded patients.

Variables	Enrolled Patients N: 89	Excluded Patients N: 29	*p*
Age (years)	69.45 ± 13.59 (24–95)	72 ± 12.59 (47–89)	*p*: 0.73
Sex (*n*)	M: 51/F: 38	M: 13/F: 16	*p*: 0.286
Diabetes mellitus (*n*, %)	17 (19.1%)	9 (31%)	*p*: 0.308
Hypertension (*n*, %)	58 (65.2%)	19 (65.5%)	*p* > 0.999
Dyslipidemia (*n*, %)	23 (25.8%)	7 (24.1%)	*p* > 0.999
Af (*n*, %)	50 (56.2%)	10 (34.5%)	*p*: 0.053
Smoking (*n*, %)	26 (29.7%)	6 (20.7%)	*p*: 0.474
Previous stroke (*n*, %)	15 (16.9%)	7 (24.1%)	*p*: 0.415
Cancer (*n*, %)	12 (13.5%)	7 (24.1%)	*p*: 0.243
Alcohol (*n*, %)	16 (18%)	4 (13.8%)	*p*: 0.778
NOACs (*n*, %)	36 (40.4%)	6 (20.7%)	* *p*: 0.048
Initial NIHSS median (IQR)	14 (10.5–17)	17 (14.5–21)	* *p*: 0.017
Pre-stroke mRS median (IQR)	0 (0–0)	0 (0–1)	*p*: 0.241
ASPECTS median (IQR)	8 (3–10.3)	6 (4.5–7)	** p*: 0.004
r-tPA (*n*, %)	28 (31.5%)	8 (27.6%)	*p*: 0.818
TICI of 2b to 3 (*n*, %)	71 (79.81%)	19 (65.5%)	* *p*: 0.002

Data are *n* (%) or means and standard deviations or medians and interquartile ranges. Af, atrial fibrillation; r-tPA, recombinant tissue-type plasminogen activator; NOAC, new oral anticoagulant; NIHSS, National Institutes of Health Stroke Score; mRS, modified Rankin Scale; ASPECTS, Alberta Stroke Program Early CT Score; TICI, Thrombolysis in Cerebral Infarction; “*”, *p* < 0.05.

**Table 2 neurolint-17-00067-t002:** Baseline characteristics and correlations with mRS score at 3 months.

Variables		Correlation with mRS Score at 3 Months
Age (years) (mean ± SD, range)	69.45 ± 13.59 (24–95)	0.296; *p*: 0.005
Sex (*n*, %)	M: 51 (57.3%)/F: 38 (42.7%)	−0.106; *p*: 0.265
Diabetes mellitus (*n*, %)	17 (19.1%)	−0.080; *p*: 0.459
Hypertension (*n*, %)	58 (65.2%)	0.044; *p*: 0.687
Dyslipidemia (*n*, %)	23 (25.8%)	0.001; *p*: 0.992
Af (*n*, %)	50 (56.2%)	0.066; *p*: 0.542
Smoking (*n*, %)	26 (29.7%)	−0.105; *p*: 0.334
Previous stroke (*n*, %)	15 (16.9%)	0.111; *p*: 0.306
Cancer (*n*, %)	12 (13.5%)	0.113; *p*: 0.290
Alcohol (*n*, %)	16 (18%)	−0.083; *p*: 0.445
NOACs (*n*, %)	36 (40.4%)	−0.026; *p*: 0.814
Initial NIHSS median (IQR)	14 (10.5–17)	0.226; *p*: 0.035
Pre-stroke mRS median (IQR)	0 (0–0)	0.200; *p*: 0.064
ASPECTS median (IQR)	8 (6–9)	−0.453; *p* < 0.001
r-tPA (*n*, %)	28 (31.5%)	−0.342; *p*: 0.001
TICI of 2b to 3 (*n*, %)	71 (79.81%)	−0.353; *p*: 0.001
QTI/non-QTI median (IQR)	0.99 (0.77–1.17)	−0.247; *p*: 0.021
QTI/QTC ratio (mean ± SD)	0.63 ± 0.16	−0.221; *p*: 0.04

Data are *n* (%) or means ± standard deviations or medians and interquartile ranges; Spearman’s correlation analysis was used to identify correlations between prognostic factors and clinical outcomes. Af, atrial fibrillation; r-tPA, recombinant tissue-type plasminogen activator; NOAC, new oral anticoagulant; NIHSS, National Institutes of Health Stroke Score; mRS, modified Rankin Scale; ASPECTS, Alberta Stroke Program Early CT Score; TICI, Thrombolysis in Cerebral Infarction; QTI/non-QTI, treatment-side internal carotid artery and non-treatment-side internal carotid artery; QTI/QTC, treatment-side internal carotid artery and treatment-side common carotid artery.

**Table 3 neurolint-17-00067-t003:** Predictors of fair outcomes in patients undergoing EVT.

Variables	mRS: 0–3 n: 53	mRS: 4–6 n: 36	*p*	Univariate Analysis OR (95% CI), *p*	Multivariate Analysis OR (95% CI), *p*
Age (years) (mean ± SD)	67.19 ± 14.21	72.78 ± 12.07	0.057	0.97 (0.93–1.00), *p*: 0.061	0.96 (0.92–1.00), *p*: 0.08
female (*n*, %)	19 (35.84%)	19 (52.77%)	0.131	2 (0.84–4.73), *p*: 0.115	1.38 (0.44–4.36), *p*: 0.58
Diabetes Mellitus (*n*, %)	10 (18.87%)	7 (19.44%)	>0.999	0.96 (0.33–2.82), *p*: 0.946	
Hypertension (*n*, %)	33 (62.26%)	25 (69.44%)	0.507	0.73 (0.29–1.79), *p*: 0.486	
Dyslipidemia (*n*, %)	12 (22.64%)	11 (30.55%)	0.464	0.67 (0.25–1.73), *p*: 0.404	
Af (*n*, %)	30 (56.60%)	20 (55.55%)	>0.999	1.04 (0.44–2.45), *p*: 0.922	
Smoking (*n*, %)	8 (15.09%)	8 (22.22%)	0.342	1.80 (0.68–4.75), *p*: 0.235	
Previous Stroke (*n*, %)	7 (13.21%)	8 (22.22%)	0.387	0.53 (0.17–1.63), *p*: 0.269	
Cancer (*n*, %)	5 (9.43%)	7 (19.44%)	0.214	0.43 (0.13–1.49), ** p*: 0.183	1.04 (0.22–4.86), *p*: 0.96
Alcohol (*n*, %)	11 (20.75)	5 (13.89)	0.575	1.62 (0.51–5.15), *p*: 0.411	
r-tPA (*n*, %)	22 (41.51)	6 (16.67)	* 0.019	3.55 (1.26–9.97), ** p*: 0.016	2.89 (0.85–9.89), *p*: 0.08
NOACs (*n*, %)	22 (41.51)	14 (38.89)	0.829	1.1 (0.47–2.56), *p*: 0.805	
Initial NIHSS median (IQR)	14 (10–17)	14 (11–20)	0.152	0.94 (0.86 -1.03), * *p*: 0.154	0.98 (0.87–1.10), *p*: 0.75
ASPECTS median (IQR)	8 (7–9)	7 (6–8)	* <0.001	1.93 (1.38–2.72), ** p* < 0.001	1.79 (1.16–2.78), *p*: 0.009
TICI of 2b to 3 (*n*, %)	48 (90.57)	23 (63.89)	* 0.003	5.43 (1.73–17.05), ** p*: 0.004	4.91 (1.10–21.89), *p*: 0.037
QTI/QTC (mean ± SD)	0.66 ± 0.13	0.57 ± 0.19	* 0.009	* 49.51 (1.99–1232.59), *p*: 0.017	45.35 (1.11–1847.51) *p*: 0.04

Data are *n* (%) or mean ± standard deviation values. * Relevant variables and key demographic indicators with a *p*-value less than 0.2 in the univariate analysis were included in the multivariate logistic regression analysis. OR, odds ratio; CI, confidence interval; Af, atrial fibrillation; r-tPA, recombinant tissue-type plasminogen activator; NOAC, new oral anticoagulant; NIHSS, National Institutes of Health Stroke Score; ASPECTS, Alberta Stroke Program Early CT Score; TICI, Thrombolysis in Cerebral Infarction; QTI/QTC, the treatment-side internal carotid artery and treatment-side common carotid artery.

**Table 4 neurolint-17-00067-t004:** Multivariate logistic regression analysis validating outcome predictors.

Variables	Odds Ratio	95% CI	*p*-Value
TICI: 2b to 3	3.72	0.97–14.27	0.056
Age < 70 y/o	2.58	0.87–7.71	0.089
ASPECTS > 8	72.89	0.94–8.91	0.064
Initial NIHSS < 15	2.00	0.65–6.15	0.225
Applying r-tPA	4.25	1.25–14.46	* 0.002
QTI/QTC > 0.6	3.83	1.28–11.48	* 0.016
Constant	0.002		0.001

OR, odds ratio; CI, confidence interval; r-tPA, recombinant tissue-type plasminogen activator; NIHSS, National Institutes of Health Stroke Score; ASPECTS, Alberta Stroke Program Early CT Score; TICI, Thrombolysis in Cerebral Infarction; QTI/QTC, the treatment-side internal carotid artery and treatment-side common carotid artery; “*”, *p* < 0.05.

## Data Availability

The data are available upon request.

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
