# Peer review of "The Association of Cerebral Blood Flow Measured Using Extracranial Carotid Ultrasound with Functional Outcomes in Patients with Anterior Circulation Large Vessel Occlusion After Endovascular Thrombectomy—A Retrospective Study"

_2035-8377, 2025, doi:10.3390/neurolint17050067_

Round 1

Reviewer 1 Report

Comments and Suggestions for Authors

This paper reports on a single-centre retrospective cohort study of the association of extracranial carotid artery blood flow study performed in the second week after endovascular thrombectomy for acute ischaemic stroke, with functional outcome. The authors analysed the data from 89 patients, and found that a higher ratio of flow of the internal vs common carotid artery was associated with a better outcome.

While this is an interesting and novel finding, there are some issues the authors may wish to address, some major, including a lack of clarity on the timing of the ultrasound, the inappropriate use of statistical analysis by correlation, totally missing information on 1 of the 2 study outcomes (mortality) stated in Aims, and the lack of sufficient discussion comparing with other studies (expanded below):

  1. Title – may I suggest an amendment to add ‘extracranial carotid’ before ‘ultrasound’, and change ‘to’ to ‘with’
  2. Lines 13-14 – while it may be true that ‘Few studies have examined the evolution of cerebral flow after the acute stage of ischemic stroke’, such ‘evolution’ was not the focus of this study
  3. Lines 15-16 – it is poorly phrased, I suggest amend to ‘we examined the association of cerebral blood flow by extracranial carotid artery sonography during the subacute phase after EVT and multiple prognostic variables, with functional outcomes’
  4. (major) Lines 19-20 – is this correct? I see in lines 243-244 that it was only performed in the 2nd week?
  5. Line 22 – please add demographic data of study subjects eg age, sex, % in the 2 outcome groups
  6. Line 37 – is ref 3 relevant? Are refs 4 and 5 already encompassed in refs 1 and 2? If yes, why repeat them here? If not, have the other landmark studies been left out?
  7. Line 46 – suggest remove ‘Consequently’
  8. Line 60 - i suggest add ‘in the second week after EVT’ after ‘sonography’ - I see in lines 243-244 that is was only performed in the 2nd week
  9. Lines 61 – please spell out ‘ICA’ and ‘CCA’ in full at first use
  10. Line 72 – suggest replace ‘resulting from’ with ‘for’
  11. (major) Line 78 – what about other factors that affect outcome such as admission blood glucose, temperature, pre-stroke functional status, etc etc?
  12. (major) Line 82 – as mentioned before, I see in lines 243-244 that it was only performed in the 2nd week – so please clarify
  13. (major) Lines 82-83 – I don’t see any data on mortality in the paper although this appears in the study Aims
  14. Line 91 – was ‘transcranial’ performed? Line 92 says ‘extracranial’
  15. (major) Lines 93-94 – a clear statement of what were measured and calculated is needed
  16. Line 99 – any additional value in calculating ratio of TI/TC ratio to nonTI/nonTC ratio?
  17. (major) Line 100 – what about mortality (mentioned in lines 82-83)
  18. (major) lines 105-108 – Pearsons is only applicable to continuous variables?
  19. Lines 108-112 - seem to overlap – please simplify into 1 sentence
  20. Lines 109, 154 – what is ‘positive outcomes’? They were not defined earlier. Are these additional study outcomes?
  21. (major) Line 123 – how do the 29 excluded subjects differ from the 89 included subjects? 25% is a high proportion of exclusions
  22. (major) Tables 1 and 2 and the related text lines 127-151 – as Pearsons is an incorrect statistical technique, these can be removed/modified
  23. Table 3 – please show female rather than male data, as per current convention. I see ‘initial mRS’ that appeared in Tables 1 and 2 is now left out - what is this entity? Pre-stroke mRS? Admission mRS? If the latter, it is surprising low…
  24. (major) Line 166 – does the result differ of all variables were included in the multivariable analysis?
  25. Lines 175-181 – these can be absorbed into the Introduction
  26. Lines 182-184 – these can be removed
  27. Line 186 – suggest adding ‘extracranial carotid’ before ‘ultrasonography’
  28. (major) Discussion – a far greater discussion is needed comparing this study findings with other studies
  29. (major) Line 188 – ref 17 is about basilar artery as is thus inappropriate. What about PRE, THRIVE-EVT, CLEAR, BET, BAND, PANDA scores?
  30. Line 217 – what is ‘LOV’?
  31. (Major) Line 217 and elsewhere in the paper – do the authors mean leptomeningeal collaterals?
  32. Line 240 – additional limitations include the exclusion of 25% of patients, lack of inclusion and analysis of other potential prognostic variables eg glucose, etc etc
Comments on the Quality of English Language

some attention needed

Author Response

Comments 1: Title – may I suggest an amendment to add ‘extracranial carotid’ before ‘ultrasound’, and change ‘to’ to ‘with’

Response 1: Thank you for pointing this out. We agree with this comment. Therefore, we have modified as instructed.

Comments 2:  Lines 13-14 – while it may be true that ‘Few studies have examined the evolution of cerebral flow after the acute stage of ischemic stroke’, such ‘evolution’ was not the focus of this study

Response 2: Thank you for pointing this out.  In this study, we used the cerebral blood flow volume obtained through sonography in the second week after stroke, a stage that represents the gradual normalization of cerebral autoregulation after the recanalization therapy, to evaluate the total effect of recanalization, cerebral autoregulation, and perfusion on surviving tissue. Therefore, we used the word “evolution”.

Comments 3:  Lines 15-16 – it is poorly phrased, I suggest amending to ‘we examined the association of cerebral blood flow by extracranial carotid artery sonography during the subacute phase after EVT and multiple prognostic variables, with functional outcomes’

Response 3: Thank you for your insightful comment. We agree with your suggestion and have revised the sentence as recommended. The updated version appears in lines 63–65.

Comments 4:  (major) Lines 19-20 – is this correct? I see in lines 243-244 that it was only performed in the 2nd week?

Response 4: Thank you for pointing this out. We agree with this comment. Therefore, we have modified as instructed. The updated version appears in lines 70–71.

Comments 5:  Line 22 – please add demographic data of study subjects eg age, sex, % in the 2 outcome groups

Response 5: Thank you for pointing this out. We agree with this comment. Therefore, we have modified as instructed. The updated version appears in lines 77.

Comments 6:  Line 37 – is ref 3 relevant? Are refs 4 and 5 already encompassed in refs 1 and 2? If yes, why repeat them here? If not, have the other landmark studies been left out?

Response 6: Thank you for pointing this out. We agree with this comment. Therefore, we have modified as instructed. The updated version appears in lines 110-111.

Comments 7:  Line 46 – suggest remove ‘Consequently’

Response 7: Thank you for pointing this out. We agree with this comment. Therefore, we have modified as instructed. The updated version appears in lines 120.

Comments 8:  Line 60 - i suggest add ‘in the second week after EVT’ after ‘sonography’ - I see in lines 243-244 that is was only performed in the 2nd week

Response 8: Thank you for pointing this out. We agree with this comment. Therefore, we have modified as instructed. The updated version appears in lines 137.

Comments 9:  Lines 61 – please spell out ‘ICA’ and ‘CCA’ in full at first use

Response 9: Thank you for pointing this out. We agree with this comment. Therefore, we have modified as instructed. The updated version appears in lines 133-139.

Comments 10:  Line 72 – suggest replace ‘resulting from’ with ‘for’

Response 10: Thank you for pointing this out. We agree with this comment. Therefore, we have modified as instructed. The updated version appears in lines 151.

Comments 11:  (major) Line 78 – what about other factors that affect outcome such as admission blood glucose, temperature, pre-stroke functional status, etc etc?

Response 11: Thank you for pointing this out. We agree with this comment. Therefore, we have modified as instructed. The updated version appears in lines 159-161.

Comments 12:  (major) Line 82 – as mentioned before, I see in lines 243-244 that it was only performed in the 2nd week – so please clarify

Response 12: Thank you for pointing this out. We agree with this comment. Therefore, we have modified as instructed. The updated version appears in lines 163-164.

Comments 13:  (major) Lines 82-83 – I don’t see any data on mortality in the paper although this appears in the study Aims

Response 13: Thank you for pointing this out. In this study, we did not analyze this portion of mortality. Therefore, we have modified as instructed. The updated version appears in lines 163-164.

Comments 14:  Line 91 – was ‘transcranial’ performed? Line 92 says ‘extracranial’

Response 14: Thank you for pointing this out. We agree with this comment. Therefore, we have modified as instructed. The updated version appears in lines 177-182.

Comments 15:  (major) Lines 93-94 – a clear statement of what were measured and calculated is needed

Response 15: Thank you for pointing this out. We agree with this comment. Therefore, we have modified as instructed. The updated version appears in lines 177-182.

Comments 16:  Line 99 – any additional value in calculating ratio of TI/TC ratio to nonTI/nonTC ratio?

Response 16: Thank you for pointing this out. We did not collect or evaluate the ratio of TI/TC ratio to nonTI/nonTC ratio; it is an interesting point of view. We would calculate and find out if it is another meaningful parameter in our future study.

Comments 17:  (major) Line 100 – what about mortality (mentioned in lines 82-83)

Response 17: Thank you for pointing this out. In this study, we did not analyze this portion of mortality. Therefore, we have modified as instructed. The updated version appears in lines 163-164.

Comments 18:  (major) lines 105-108 – Pearsons is only applicable to continuous variables?

Response 18: Thank you for pointing this out. We agree with this comment. Therefore, we have modified as instructed. The updated version appears in lines 194-196.

Comments 19:  Lines 108-112 - seem to overlap – please simplify into 1 sentence

Response 19: Thank you for pointing this out. In this section, we tried to explain the way we selected the important factors (p<0.2 ) and then process these factors to multivariate regression. It seems to overlap, but it is a necessary clarification.  

Comments 20:  Lines 109, 154 – what is ‘positive outcomes’? They were not defined earlier. Are these additional study outcomes?

Response 20: Thank you for pointing this out. We agree with this comment. Therefore, we have modified as instructed. The updated version appears in lines 196-198, 246.

Comments 21:  (major) Line 123 – how do the 29 excluded subjects differ from the 89 included subjects? 25% is a high proportion of exclusions

Response 21: Thank you for pointing this out. We agree with this comment. Therefore, we have modified as instructed. The updated version appears in figure 1.

Comments 22:  (major) Tables 1 and 2 and the related text lines 127-151 – as Pearsons is an incorrect statistical technique, these can be removed/modified

Response 22: Thank you for pointing this out. We agree with your comment and have revised the content accordingly.

Comments 23:  Table 3 – please show female rather than male data, as per current convention. I see ‘initial mRS’ that appeared in Tables 1 and 2 is now left out - what is this entity? Pre-stroke mRS? Admission mRS? If the latter, it is surprising low…

Response 23: Thank you for pointing this out. We agree with your comment and have revised the content accordingly. The updated version appears in table 1 and 2. Due to the guideline of EVT, they recommended good candidate for treatment is the one with pre-stroke mRS ≤1. Most of our cases followed this guideline ; therefore, it is surprisingly low.

Comments 24:  (major) Line 166 – does the result differ of all variables were included in the multivariable analysis?

Response 24: Thank you for pointing this out. Because this is a small sample size study, the risk of overfitting would be higher if we included too many variables in the regression analysis. 

Comments 25:  Lines 175-181 – these can be absorbed into the Introduction

Response 25: Thank you for pointing this out. We agree with this comment. Therefore, we have modified as instructed. The updated version appears in lines 272-276.

Comments 26:  Lines 182-184 – these can be removed

Response 26: Thank you for pointing this out. We agree with this comment. Therefore, we have modified as instructed. We sincerely appreciate your helpful feedback.

Comments 27:  Line 186 – suggest adding ‘extracranial carotid’ before ‘ultrasonography’

Response 27: Thank you for pointing this out. We agree with this comment. Therefore, we have modified as instructed. The updated version appears in lines 282-283.

Comments 28:  (major) Discussion – a far greater discussion is needed comparing this study findings with other studies

Response 28: Thank you for pointing this out. We agree with this comment. Therefore, we have modified as instructed.

Comments 29:  (major) Line 188 – ref 17 is about basilar artery as is thus inappropriate. What about PRE, THRIVE-EVT, CLEAR, BET, BAND, PANDA scores?

Response 29: Thank you for pointing this out and we replaced this reference , which fits the topic of anterior circulation . Totaled Health Risks in Vascular Events (THRIVE) score has been shown to predict poor functional outcome in patients with acute ischemic stroke (AIS) and anterior circulation large vessel occlusions undergoing thrombectomy treatment. The updated version appears in lines 284-286.

Comments 30:  Line 217 – what is ‘LOV’?

Response 30: Thank you for pointing this out. We agree with this comment. Therefore, we have modified as instructed. The updated version appears in lines 333.

Comments 31:  (Major) Line 217 and elsewhere in the paper – do the authors mean leptomeningeal collaterals?

Response 31: The collateral flow means the flow from ipsilateral ECA (including leptomeningeal or reversed ophthalmic artery) or circle of Willis (Acom or contralateral ACA or ipsilateral PCA )

Comments 32:  Line 240 – additional limitations include the exclusion of 25% of patients, lack of inclusion and analysis of other potential prognostic variables eg glucose, etc etc

Response 32: Thank you for pointing this out. We agree with this comment. Therefore, we have modified as instructed. The updated version appears in lines 366-375.

Reviewer 2 Report

Comments and Suggestions for Authors

Title

“Cerebral Blood Flow Measured by Ultrasound”

Line 248

“metric to identify patients who are most likely to benefit from EVT”

Line 243

“sonography was performed starting from day7 to day 14 after EVT”

Line 252

“several relevant factors of patient outcomes”

“age, r-tPA administration, initial mRS score, ASPECTS, recanalization success,”

COMMENT

Line 248 offers to help clinicians to decide whether EVT is appropriate therapy for a patient presenting to the emergency department with symptoms of ischemic stroke.  At the moment of evaluation, the clinician could know the age of the patient (Table 1 p=0.006).  The clinician can assess stroke scores like mRS (able to walk, Table 1 p=0.013), and can take the patient to an imaging suite for a non-contrast brain CT scan for ASPECTS scoring (Table 1 p<0.001), and a contrast CT scan to rule out hemorrhagic stroke and establish a TICI evaluation (Table 1 p=0.001).  Following the successful EVT, the elective use of “r-tPA” (Table 1 p=0.001) is also an important way to improve outcome.  The TI/TC score (Table 1 p=0.011) used in this analysis and headlined in the title “Cerebral Blood Flow Measured by Ultrasound” is available only 2 weeks after the key clinical decisions have been made. 

In reference 16 (Kim), the time from symptom onset to perfusion imaging is about 3 hours, not 2 weeks.

Reference 16 says “collateral status before treatment is also an important determinant of tissue fate.”

In reference 14 (Villringer) thrombolysis might have been used or just spontaneous re-canulation monitored.  Both reference 14 and 16 refer to MRI perfusion imaging of the brain.

The “TI/TC” score used in this manuscript is not really a measure of blood flow through the brain.

Line 245

“we did not exclude patients with hemorrhagic conversion”

COMMENT

Because of the importance of “r-tPA” in the post-EVA treatment, and because hemorrhagic conversion would likely be reported in the hospital records of the 89/1789 patients evaluated in this report, the incidence of hemorrhagic conversion in cases receiving and not receiving post-TVA “r-tPA” should be included in the analysis.

Line 25

“Thrombolysis in cerebral infarction 2b to 3 (OR 4.91; 95%CI 1.10-21.89; p=0.037)”

Line 27

“ratio of treatment-side blood flow between internal carotid artery and common carotid artery (TI/TC, OR 45.35; 95%CI 1.11-1847.51 ; p=0.04)”

COMMENT

“Thrombolysis in cerebral infarction…” is a binary “yes/no” variable.

I think that “fair outcomes” (Line 27) is a binary variable.

OR can be computed from the relationship between two binary variables.

BUT “TI/TC” on the ipsilateral side and the contralateral side are both continuous variables; the method of computing an OR of 45.35 from continuous variables is not obvious

Line 89

“Recanalization success was evaluated using the Thrombolysis in Cerebral Infarction (TICI) grading system on the final control angiogram.”

COMMENT

Although GOOGLE was able to find an explanation of the TICI score and the difference between 2b and 3, the authors should provide a key reference for each of the methods used in this report for the reader that is not engaged in reperfusion work.

Ref 14 and 16

  1. Villringer K, Zimny S, Galinovic I, Nolte CH, Fiebach JB, Khalil AA. The Association Between Recanalization, Collateral Flow,
  2. Kim SJ, Son JP, Ryoo S, Lee MJ, Cha J, Kim KH, et al. A novel magnetic resonance imaging approach to collateral flow

This manuscript uses variable “TI/TC”

Line 27

(TI/TC, OR 45.35; 95%CI 1.11-1847.51 ; p=0.04)

Line 97

treatment-side ICA and treatment-side CCA(abbreviated as the TI/TC ratio) and between the treatment-side ICA and non-treatment-side ICA(abbreviated as the TI/non-TI ratio).

Line 155

“positive outcomes”  “high TI/TC ratios(Table 3)” 

((0.66 è good outcome; 0.57 è [poor outcome (mRS:4-6)

Line 55

“Cerebral sonography facilitates the real-time assessment of cerebral blood flow”

Line 81

“All patients underwent carotid sonography between the first to the secondary weeks after EVT”

COMMENT

The ultrasound examination is not described in the manuscript.  The reason for the carotid ultrasound examination is not described in the manuscript.  Neither “TI” nor “TC” nor “TI/TC” are commonly used terms in reporting clinical carotid ultrasound Doppler examinations.  Two references are provided to support the carotid examination methods 26 (Scheel) and 27 (Schoning). 

Line 223

In healthy adults, TI/TC ranged between 0.58 to 0.65. [26] [27].

COMMENT CONTINUED

Neither publication uses the terms TI or TC; Scheel refers to “time-averaged flow velocity (TAV)”; Schoning uses the terms “TAV, time-averaged velocity; TAMX, time-averaged maximum velocity;” neither mentions a ratio.  In conventional clinical diagnostic carotid Doppler duplex scanning, a ICAPSV/CCAPSV ratio has been used where ICA refers to the internal carotid artery, CCA refers to the common carotid artery and PSV refers to the measured peak systolic velocity.  Schoning and Scheel are colleagues from Tubingen, Germany.  The method of computing volume flow in peripheral arteries from Doppler velocity measurements and vessel dimensions has not been generally adopted as a standard of practice elsewhere.  This strategy has been commonly used to measure cardiac output when the ultrasound Doppler examination angle from the super-sternal notch to the aortic outflow tract axis allows a Doppler angle of ZERO and a wide ultrasound beam is used to encompass the entire aortic cross section so that the average Time-Average Velocity measurement is equal to the flow progression of blood perpendicular to the computed cross section.  In peripheral arteries, with a Doppler angle near 60 degrees interrogating helical flow, the simplifying assumptions are not valid.  For instance, in the Schoning paper, the average CCA flow (470 [mL/minute]) entering the carotid bifurcation should equal the sum of the ICA and ECA flows (265 [mL/min] + 160 [mL/min] = 425 [mL/min]).  The likely reason for the difference is that the assumptions about flow used in the Doppler method are not valid.

The present manuscript does not describe why the Doppler examinations were done or how they were done.  If they are standard clinical examinations, then the measurements used for TI and TC are not similar to the Time Average measurements described in the references.

ABOVE ARE COMMENTS FROM THE SECOND READING OF THE PAPER

BELOW ARE COMMENTS FROM THE FIRST DAY SPENT READING AND TRYING TO UNDERSTAND THIS PAPER.

UPPER CASE LETTERS ARE USED IN THIS REVIEW TO INDICATE COMMENTS MADE BY THE REVIEWER.

Lower case letters are used to indicate text from the manuscript.

Title

“Cerebral Blood Flow Measured by Ultrasound”

“CEREBRAL BLOOD FLOW” IS GENERALLY INTERPRETED TO MEAN CUBIC CENTIMETERS PER SECOND (OR MILLILITERS PER SECOND) (OR PER MINUTE) THROUGH THE BRAIN.  VOLUME FLOW (USUALLY CALLED “Q”) CAN BE COMPUTED BY MULTIPLYING THE VELOCITY TIMES THE CROSS-SECTIONAL AREA OF THE ARTERY.  TO MAKE SUCH A MEASUREMENT WITH ULTRASOUND WOULD REQUIRE MEASUREMENT OF THE AVERAGE VELOCITY IN AN ARTERY OR VEIN AND MEASUREMENT OF CROSS-SECTIONAL AREA.  IF THE VESSEL HAS A CIRCULAR CROSS SECTION, THEN DIAMETER CAN BE USED TO COMPUTE CROSS-SECTION.  ALTHOUGH DOPPLER STUDIES OF THE CAROTID ARTERIES ARE OFTEN REPORTED AS VELOCITY (“V”), THE VALUES PROVIDED ARE COMPUTED FROM THE “DOPPLER EQUATION” WHICH INCLUDES A MEASURE OF THE “DOPPLER ANGLE” WHICH IS OFTEN 60 DEGREES.  BLOOD FLOW THROUGH THE CURVED CAROTID ARTERIES IS USUALLY HELICAL, SO THE USE OF THE DOPPLER EQUATION IS PROBLEMATIC. 

THE MIDDLE CEREBRAL ARTERY (MCA) IS ALIGNED WITH THE “TEMPORAL (ULTRASOUND) WINDOW” OF THE SKULL BONE SO THAT THE “DOPPLER ANGLE” IN THE “DOPPLER EQUATION” FOR MCA VELOCITY IS ZERO AND THE MEASUREMENT CAN BE USED FOR “AVERAGE PARA-AXIAL VELOCITY” IF THE MCA DIAMETER WERE KNOWN.

PERHAPS THE TITLE SHOULD BE:

“Association of Ischemic Stroke Post-Thrombectomy Extracranial Carotid Artery Doppler Ultrasound Velocity Measurements and Other Factors with 90 Functional Outcome”

Page 1 Line 18

Patients with acute stroke resulting from anterior-circulation large vessel occlusion and underwent EVT were included.

DO YOU MEAN

Patients with acute stroke resulting from anterior-circulation large vessel occlusion who underwent EVT were included.

CONFUSING

A multivariable logistic regression analysis revealed that Alberta Stroke Program Early CT Score (odds ratio [OR] 1.79; 95% confidence interval [CI] 1.16-2.78; p=0.009), Thrombolysis in cerebral infarction 2b to 3 (OR 4.91; 95%CI 1.10-21.89; p=0.037) and the ratio of treatment-side blood flow between internal carotid artery and common carotid artery (TI/TC, OR 45.35; 95%CI 1.11-1847.51 ; p=0.04) were independent predictors of fair outcomes.

PERHAPS

A multivariable logistic regression analysis revealed that 3 factors were independent predictors of fair outcomes: 1) Alberta Stroke Program Early CT Score (odds ratio [OR] 1.79; 95% confidence interval [CI] 1.16-2.78; p=0.009); 2) Thrombolysis in cerebral infarction 2b to 3 (OR 4.91; 95%CI 1.10-21.89; p=0.037); and 3) the ratio of treatment-side blood flow between internal carotid artery and common carotid artery (TI/TC, OR 45.35; 95%CI 1.11-1847.51 ; p=0.04).

Page 1 Line 27

(TI/TC, OR 45.35; 95%CI 1.11-1847.51 ; p=0.04)

ON LINE 19

All patients underwent carotid sonography within 2 weeks after EVT.

SO THE SONOGRAPHY WAS DONE AFTER THE EVT.

USUALLY WHEN INTRODUCING AN ABBREVIATION, (TI/TC), THE ABBREVIATION IS BRACKETED WITHOUT ADDITIONAL INFORMATION.  THIS READER USED SOME EFFORT TO FIGURE OUT THE MEANING OF TI/TC, PERHAPS TI MEANS SOME INTERNAL CAROTID ARTERY FLOW PARAMETER.  IS THIS FLOW RATE [CC/SECOND], SYSTOLIC VELOCITY [CM/SECOND], DIASTOLIC VELOCITY [CM/SECOND].  ALTHOUGH THE USE OF “I” AND “C” ARE COMPREHENSIBLE FOR IDENTIFYING THE INTERNAL CAROTID ARTERY AND COMMON CAROTID ARTERY, “T” IS NOT A COMMONLY USED TERM: “V” IS COMMONLY USED FOR VELOCITY [CM/SECOND], “PSV” IS COMMONLY USED FOR PEAK SYSTOLIC VELOCITY, “Q” IS COMMONLY USED FOR FLOW RATE [CC/SECOND] OR [CC/MINUTE].

Line 63 anterior circulation large vessel occlusion (AC-LVO)

This is distal ICA, m1MCA and m2MCA, not any ACA.  I would have included ACA as part of the “anterior circulation”

Line 75

All patients had completed National Institutes of Health Stroke Score(NIHSS) evaluation , initial and 90-day modified Rankin Scale (mRS) assessments, Alberta Stroke Program Early CT Score (ASPECTS),and CT angiography (CTA)assessments before EVT.

GOOGLE SAYS

The 90-day modified Rankin Scale (mRS) is a widely used outcome measure in stroke clinical trials and clinical practice, assessing functional neurological disability after stroke at 90 days, ranging from 0 (no symptoms) to 6 (death). 

BUT THE TEXT IMPLIES THAT THE 90 DAY “MRS” WAS CONDUCTED BEFORE “EVT”.  DOES THAT MEAN THAT THE “EVT” WAS PERFORMED 90 DAYS AFTER THE STROKE?

MAYBE YOU MEAN

All patients had completed National Institutes of Health Stroke Score (NIHSS) evaluation , modified Rankin Scale (mRS) assessments, Alberta Stroke Program Early CT Score (ASPECTS),and CT angiography (CTA)assessments before EVT.  All patients were also evaluated with the 90-day modified Rankin Scale (mRS) and these other tests at 90 days after the EVT.

Line 75

All patients had completed National Institutes of Health Stroke Score (NIHSS) evaluation , initial and 90-day modified Rankin Scale (mRS) assessments, Alberta Stroke Program Early CT Score (ASPECTS),and CT angiography (CTA)assessments before EVT.

WHENEVER AN EVALUATION METHOD IS INTRODUCED (LIKE NIHSS OR ASPECTS), PLEASE CITE THE BEST REFERENCE THAT DESCRIBES THE METHOD USED SO THAT A READER HAS ENOUGH INFORMATION TO DUPLICATE THE METHOD.

Line 81

All patients underwent carotid sonography between the first to the second weeks after EVT

Line 96

To analyze the CBF in the treatment artery ….  blood flow

PERHAPS USING SONOGRAPHY THE VELOCITY WAS MEASURED IN THE DISTAL COMMON CAROTID ARTERY AND THE PROXIMAL INTERNAL CAROTID ARTERY IN THE NECK.  BUT IF THE “treatment artery” IS THE m2MCA, THEN THE MEASUREMENT IS DONE ON THE SIDE OF TREATMENT, NOT “in the treatment artery”.

TABLES

Variables in the tables are listed in an order.  The choice of the order is unclear, it is not based on R value (Pearson correlation), or on “p” values.  How was this order selected?

Line 175

“selection for potentially good candidates for EVT”

THE PAPER DOES NOT DISCUSS HOW CASES WERE SELECTED FOR EVT.  IT LOOKS LIKE ONLY 10% OF CASES RECEIVED EVT AND OF THOSE, ABOUT HALF WERE INCLUDED IN THIS ANALYSIS. (Figure 1).

IT IS UNCLEAR FROM FIGURE 1 WHEN VARIOUS VARIABLES WERE MEASURED.

IN FIGURE 1, 172 PATIENTS WITH STROKE HAD EVT, 118 HAD OCCLUSION OF THE ICA OR MCA.

WHAT WAS THE SITUATION OF THE OTHER 54 PATIENTS?

HOW WAS THEIR SITUATION DISOVERED?

WHAT WAS THE PATHOLOGY OF THE MATERIAL RETRIEVED WITH EVT?

Line 175

carefully selection for potentially good candidates

I THINK THAT YOU MEAN

Carful selection of potentially good candidates

Line 182

we investigated multiple prognostic factors, including premorbid characteristics,

GOOGLE SAYS

Premorbid characteristics refer to an individual's functioning and personality traits before the onset of a disease or illness, particularly in the context of mental health conditions. 

I SUPPOSE THAT SEX (MALE), DIABETES AND SMOKING ARE PREMORBID AND THAT YOU DISCOVERED THESE IN THE MEDICAL CHART REVIEWS FOR DATES PRIOR TO THE STROKE EPISODE, BUT “r-tPA” is not a “premorbid condition”. 

Table 3

“Old stroke” MIGHT BETTER BE CALLED “Previous stroke”

ASPECTS and other diagnostic methods are not cited in the references.  This reviewer found

Stroke. 2017;48:1574-1579. DOI: 10.1161/STROKEAHA.117.016745 to learn about the head Computed Axial Tomography (also known as CT and CAT and EMI scan) method. 

Hmmm, GOOGLE is incorrect about EMI scan.  GOOGLE says

An EMI scan, which stands for ElectroMagnetic Interference, is a type of CT (Computed Tomography) scan developed by the company EMI, and it was the first to be adopted in substantial numbers for medicine, allowing for detailed pictures of the brain. 

BUT this is not correct.

the Beatles with their record company, EMI, indirectly led to the funding of the research that developed the CT scanner, also known as a CAT scan, by EMI engineer Godfrey Hounsfield

AND

Journal of Neurology, Neurosurgery, and Psychiatry, 1975, 38, 935-947

Mokin M, Primiani CT, Siddiqui AH, Turk AS, ASPECTS (Alberta Stroke Program Early CT Score) Measurement Using Hounsfield Unit Values When Selecting Patients for Stroke Thrombectomy,

(Stroke. 2017;48:1574-1579. DOI: 10.1161/STROKEAHA.117.016745.)

USES Hounsfield unit (HU) values on initial noncontrast head computerized tomography (CT) correlates with the extent of final infarct on follow-up imaging.

Line 214

we used the cerebral blood flow volume

Line 218

We regarded the flow ratio between the ICA and CCA as the flow required from the treated territory in the ipsilateral cerebral artery.

IN A STANDARD ULTRASONIC DOPPLER EXAMINATION OF THE CAROTID BIFURCATION, VELOCITIES (NOT FLOW RATES [VOLUME/TIME] OR VOLUME) ARE MEASURED (ESTIMATED USING THE DOPPLER EQUATION) FROM THE COMMON CAROTID ARTERY, INTERNAL CAROTID ARTERY AND EXTERNAL CAROTID ARTERY.  ALTHOUGH IT IS EXPECTED THAT THE FLOW THROUGH THE COMMON CAROTID ARTERY SHOULD BE EQUAL TO THE SUM OF THE FLOW THROUGH THE INTERNAL CAROTID ARTERY AND THE EXTERNAL CAROTID ARTERY, I DON’T THINK THAT THIS “CONSERVATION OF FLOW” HAS BEEN VERIFIED USING SONOGRAPHIC METHODS, OR ANY OTHER METHOD FOR THAT MATTER.  THE FLOW RATIO “TI/TC” MIGHT BE AN INDICATION OF FLOW DIVERSION THROUGH THE EXTERNAL CAROTID ARTERY AND ONTO THE BRAIN VIA THE OPHTHALMIC ARTERY.

+++++++++++++++

Good collateral status [5-8]

Line 198

After vessel recanalization, 3 conditions may develop. [12, 18, 19] The first condition is persistent stenosis or occlusion,

In healthy adults, TI/TC 222 ranged between 0.58 to 0.65. [26] [27]

  1. Scheel P, Ruge C, Schoning M. Flow velocity and flow volume measurements in the extracranial carotid and vertebral arteries in healthy adults: reference data and the effects of age. Ultrasound Med Biol. 2000;26(8):1261-1266. doi: 10.1016/s0301-5629(00)00293-3. PubMed PMID: 11120363.

  1. Schoning M, Walter J, Scheel P. Estimation of cerebral blood flow through color duplex sonography of the carotid and vertebral arteries in healthy adults. Stroke. 1994;25(1):17-22. doi: 10.1161/01.str.25.1.17. PubMed PMID: 8266366.

Flow velocity and flow volume measurements in the extracranial carotid and vertebral arteries in healthy adults: reference data and the effects of age.

Scheel P, Ruge C, Schöning M.

Ultrasound Med Biol. 2000 Oct;26(8):1261-6. doi: 10.1016/s0301-5629(00)00293-3.

PMID: 11120363

Color duplex measurement of cerebral blood flow volume in healthy adults.

Scheel P, Ruge C, Petruch UR, Schöning M.

Stroke. 2000 Jan;31(1):147-50. doi: 10.1161/01.str.31.1.147.

PMID: 10625730

  1. Schoning M,

4-8,9,10,11

CCA diametric pulsatility

TAV is time averaged velocity

Comments on the Quality of English Language

The authors have a few odd use of words and grammar that should be corrected by the journal editors before the manuscript is sent out for scientific review.

Author Response

Reviewer 2

Comment 1: Line 248 offers to help clinicians to decide whether EVT is appropriate therapy for a patient presenting to the emergency department with symptoms of ischemic stroke.  At the moment of evaluation, the clinician could know the age of the patient (Table 1 p=0.006).  The clinician can assess stroke scores like mRS (able to walk, Table 1 p=0.013), and can take the patient to an imaging suite for a non-contrast brain CT scan for ASPECTS scoring (Table 1 p<0.001), and a contrast CT scan to rule out hemorrhagic stroke and establish a TICI evaluation (Table 1 p=0.001).  Following the successful EVT, the elective use of “r-tPA” (Table 1 p=0.001) is also an important way to improve outcome.  The TI/TC score (Table 1 p=0.011) used in this analysis and headlined in the title “Cerebral Blood Flow Measured by Ultrasound” is available only 2 weeks after the key clinical decisions have been made. 

Response 1: Thank you for pointing this out. We agree with this comment. Therefore, we have modified as instructed. The updated version appears in lines 291-305,  329-344  and  375–377.

Comment 2:

In reference 16 (Kim), the time from symptom onset to perfusion imaging is about 3 hours, not 2 weeks.

Reference 16 says “collateral status before treatment is also an important determinant of tissue fate.”

In reference 14 (Villringer) thrombolysis might have been used or just spontaneous re-canulation monitored.  Both reference 14 and 16 refer to MRI perfusion imaging of the brain.

The “TI/TC” score used in this manuscript is not really a measure of blood flow through the brain.

Ref 14 and 16

  1. Villringer K, Zimny S, Galinovic I, Nolte CH, Fiebach JB, Khalil AA. The Association Between Recanalization, Collateral Flow,
  2. Kim SJ, Son JP, Ryoo S, Lee MJ, Cha J, Kim KH, et al. A novel magnetic resonance imaging approach to collateral flow

Response 2: Thank you for pointing this out. We agree with this comment. Therefore, we have explained and further clarified  it . The updated version appears in lines 343-365 .

Comment 3:

Line 245

“we did not exclude patients with hemorrhagic conversion”

COMMENT

Because of the importance of “r-tPA” in the post-EVA treatment, and because hemorrhagic conversion would likely be reported in the hospital records of the 89/1789 patients evaluated in this report, the incidence of hemorrhagic conversion in cases receiving and not receiving post-TVA “r-tPA” should be included in the analysis.

Response 3: Thank you very much for your valuable comment. We understand the importance of considering hemorrhagic conversion, particularly in relation to post-EVT administration of r-tPA. However, in the present study, we did not perform a detailed analysis of hemorrhagic conversion events as this information was not consistently available in our dataset. We acknowledge this as a limitation and have now noted it accordingly in the discussion section. Thank you again for your insightful suggestion. The updated version appears in lines 371–372.

Comment 4:

Line 25

“Thrombolysis in cerebral infarction 2b to 3 (OR 4.91; 95%CI 1.10-21.89; p=0.037)”

Line 27

“ratio of treatment-side blood flow between internal carotid artery and common carotid artery (TI/TC, OR 45.35; 95%CI 1.11-1847.51 ; p=0.04)”

COMMENT

“Thrombolysis in cerebral infarction…” is a binary “yes/no” variable.

I think that “fair outcomes” (Line 27) is a binary variable.

OR can be computed from the relationship between two binary variables.

BUT “TI/TC” on the ipsilateral side and the contralateral side are both continuous variables; the method of computing an OR of 45.35 from continuous variables is not obvious

Response 4: Thank you for pointing this out. We agree with this comment. Because this study is  small sample size and included multiple variables in the regression analysis .It  increases the risk of overfitting. This could explain the implausibly large OR for the TI/TC ratio. Therefore, we have modified as instructed . We acknowledge this as a limitation and have now noted it accordingly in the discussion section .The updated version appears in lines 373-375.

Comment 5:

Line 89

“Recanalization success was evaluated using the Thrombolysis in Cerebral Infarction (TICI) grading system on the final control angiogram.”

COMMENT

Although GOOGLE was able to find an explanation of the TICI score and the difference between 2b and 3, the authors should provide a key reference for each of the methods used in this report for the reader that is not engaged in reperfusion work.

Response 5: Thank you for pointing this out. We agree with this comment. Therefore, we have modified as instructed. The updated version appears in lines 172–176.

Comment 6: 

This manuscript uses variable “TI/TC”

Line 27

(TI/TC, OR 45.35; 95%CI 1.11-1847.51 ; p=0.04)

Line 97

treatment-side ICA and treatment-side CCA(abbreviated as the TI/TC ratio) and between the treatment-side ICA and non-treatment-side ICA(abbreviated as the TI/non-TI ratio).

Line 155

“positive outcomes”  “high TI/TC ratios(Table 3)” 

((0.66 è good outcome; 0.57 è [poor outcome (mRS:4-6)

Line 55

“Cerebral sonography facilitates the real-time assessment of cerebral blood flow”

Line 81

“All patients underwent carotid sonography between the first to the secondary weeks after EVT”

COMMENT

The ultrasound examination is not described in the manuscript.  The reason for the carotid ultrasound examination is not described in the manuscript.  Neither “TI” nor “TC” nor “TI/TC” are commonly used terms in reporting clinical carotid ultrasound Doppler examinations.  Two references are provided to support the carotid examination methods 26 (Scheel) and 27 (Schoning). 

Line 223

In healthy adults, TI/TC ranged between 0.58 to 0.65. [26] [27].

COMMENT CONTINUED

Neither publication uses the terms TI or TC; Scheel refers to “time-averaged flow velocity (TAV)”; Schoning uses the terms “TAV, time-averaged velocity; TAMX, time-averaged maximum velocity;” neither mentions a ratio.  In conventional clinical diagnostic carotid Doppler duplex scanning, a ICAPSV/CCAPSV ratio has been used where ICA refers to the internal carotid artery, CCA refers to the common carotid artery and PSV refers to the measured peak systolic velocity.  Schoning and Scheel are colleagues from Tubingen, Germany.  The method of computing volume flow in peripheral arteries from Doppler velocity measurements and vessel dimensions has not been generally adopted as a standard of practice elsewhere.  This strategy has been commonly used to measure cardiac output when the ultrasound Doppler examination angle from the super-sternal notch to the aortic outflow tract axis allows a Doppler angle of ZERO and a wide ultrasound beam is used to encompass the entire aortic cross section so that the average Time-Average Velocity measurement is equal to the flow progression of blood perpendicular to the computed cross section.  In peripheral arteries, with a Doppler angle near 60 degrees interrogating helical flow, the simplifying assumptions are not valid.  For instance, in the Schoning paper, the average CCA flow (470 [mL/minute]) entering the carotid bifurcation should equal the sum of the ICA and ECA flows (265 [mL/min] + 160 [mL/min] = 425 [mL/min]).  The likely reason for the difference is that the assumptions about flow used in the Doppler method are not valid.

The present manuscript does not describe why the Doppler examinations were done or how they were done.  If they are standard clinical examinations, then the measurements used for TI and TC are not similar to the Time Average measurements described in the references.

 Response 6: Thank you for  those important points.

Firstly , we added the method and key parameters of extracranial sonography in line  178-182.Secondly , we explained why we targeted the FV of CCA and ICA in line 133-140. The method of computing volume flow in peripheral arteries from Doppler velocity measurements and vessel dimensions has  been generally adopted as a standard of exam in our daily practice. And ,after the improvement of sonography machine, the sum FV of ICA and ECA usually could match the CCA better than before . The TI/TC and TI/non-TI  ratios  are a novel index,  and we are pleased  to present and discuss these findings.

Comment 7: 

Title

“Cerebral Blood Flow Measured by Ultrasound”

“CEREBRAL BLOOD FLOW” IS GENERALLY INTERPRETED TO MEAN CUBIC CENTIMETERS PER SECOND (OR MILLILITERS PER SECOND) (OR PER MINUTE) THROUGH THE BRAIN.  VOLUME FLOW (USUALLY CALLED “Q”) CAN BE COMPUTED BY MULTIPLYING THE VELOCITY TIMES THE CROSS-SECTIONAL AREA OF THE ARTERY.  TO MAKE SUCH A MEASUREMENT WITH ULTRASOUND WOULD REQUIRE MEASUREMENT OF THE AVERAGE VELOCITY IN AN ARTERY OR VEIN AND MEASUREMENT OF CROSS-SECTIONAL AREA.  IF THE VESSEL HAS A CIRCULAR CROSS SECTION, THEN DIAMETER CAN BE USED TO COMPUTE CROSS-SECTION.  ALTHOUGH DOPPLER STUDIES OF THE CAROTID ARTERIES ARE OFTEN REPORTED AS VELOCITY (“V”), THE VALUES PROVIDED ARE COMPUTED FROM THE “DOPPLER EQUATION” WHICH INCLUDES A MEASURE OF THE “DOPPLER ANGLE” WHICH IS OFTEN 60 DEGREES.  BLOOD FLOW THROUGH THE CURVED CAROTID ARTERIES IS USUALLY HELICAL, SO THE USE OF THE DOPPLER EQUATION IS PROBLEMATIC. 

THE MIDDLE CEREBRAL ARTERY (MCA) IS ALIGNED WITH THE “TEMPORAL (ULTRASOUND) WINDOW” OF THE SKULL BONE SO THAT THE “DOPPLER ANGLE” IN THE “DOPPLER EQUATION” FOR MCA VELOCITY IS ZERO AND THE MEASUREMENT CAN BE USED FOR “AVERAGE PARA-AXIAL VELOCITY” IF THE MCA DIAMETER WERE KNOWN.

PERHAPS THE TITLE SHOULD BE:

“Association of Ischemic Stroke Post-Thrombectomy Extracranial Carotid Artery Doppler Ultrasound Velocity Measurements and Other Factors with 90 Functional Outcome”

 Response 7: Thank you for pointing this out .

In reference 35 , “Schoning M, Walter J, Scheel P. Estimation of cerebral blood flow through color duplex sonography of the carotid and vertebral arteries in healthy adults. Stroke. 1994  presented the details of cerebral blood flow measured by ultrasound . The use of the doppler equation is not so problematic in our daily practice. About the difference and clinical application between velocity and FV , we added it in discussion .The updated version appears in lines 291-314.

Comment 8:

Page 1 Line 18

Patients with acute stroke resulting from anterior-circulation large vessel occlusion and underwent EVT were included.

DO YOU MEAN

Patients with acute stroke resulting from anterior-circulation large vessel occlusion who underwent EVT were included.

Response 8: Thank you for pointing this out. We agree with this comment. Therefore, we have modified as instructed. The updated version appears in lines 69–71.

Comment 9:

CONFUSING

A multivariable logistic regression analysis revealed that Alberta Stroke Program Early CT Score (odds ratio [OR] 1.79; 95% confidence interval [CI] 1.16-2.78; p=0.009), Thrombolysis in cerebral infarction 2b to 3 (OR 4.91; 95%CI 1.10-21.89; p=0.037) and the ratio of treatment-side blood flow between internal carotid artery and common carotid artery (TI/TC, OR 45.35; 95%CI 1.11-1847.51 ; p=0.04) were independent predictors of fair outcomes.

PERHAPS

A multivariable logistic regression analysis revealed that 3 factors were independent predictors of fair outcomes: 1) Alberta Stroke Program Early CT Score (odds ratio [OR] 1.79; 95% confidence interval [CI] 1.16-2.78; p=0.009); 2) Thrombolysis in cerebral infarction 2b to 3 (OR 4.91; 95%CI 1.10-21.89; p=0.037); and 3) the ratio of treatment-side blood flow between internal carotid artery and common carotid artery (TI/TC, OR 45.35; 95%CI 1.11-1847.51 ; p=0.04).

Response 9: Thank you for pointing this out. We agree with this comment. Therefore, we have modified as instructed. The updated version appears in lines 77–82.

Comment 10:  

Page 1 Line 27

(TI/TC, OR 45.35; 95%CI 1.11-1847.51 ; p=0.04)

ON LINE 19

All patients underwent carotid sonography within 2 weeks after EVT.

SO THE SONOGRAPHY WAS DONE AFTER THE EVT.

USUALLY WHEN INTRODUCING AN ABBREVIATION, (TI/TC), THE ABBREVIATION IS BRACKETED WITHOUT ADDITIONAL INFORMATION.  THIS READER USED SOME EFFORT TO FIGURE OUT THE MEANING OF TI/TC, PERHAPS TI MEANS SOME INTERNAL CAROTID ARTERY FLOW PARAMETER.  IS THIS FLOW RATE [CC/SECOND], SYSTOLIC VELOCITY [CM/SECOND], DIASTOLIC VELOCITY [CM/SECOND].  ALTHOUGH THE USE OF “I” AND “C” ARE COMPREHENSIBLE FOR IDENTIFYING THE INTERNAL CAROTID ARTERY AND COMMON CAROTID ARTERY, “T” IS NOT A COMMONLY USED TERM: “V” IS COMMONLY USED FOR VELOCITY [CM/SECOND], “PSV” IS COMMONLY USED FOR PEAK SYSTOLIC VELOCITY, “Q” IS COMMONLY USED FOR FLOW RATE [CC/SECOND] OR [CC/MINUTE].

Response 10: Thank you for pointing this out. We agree with this comment. Therefore, we have modified as instructed. The updated version appears in lines 185–187.

Comment 11:  

Line 63 anterior circulation large vessel occlusion (AC-LVO)

This is distal ICA, m1MCA and m2MCA, not any ACA.  I would have included ACA as part of the “anterior circulation”

Response 11: Thank you for pointing this out. We agree with this comment. Therefore, we have modified as instructed. The updated version appears in figure 1.

Comment 12:

Line 75

All patients had completed National Institutes of Health Stroke Score(NIHSS) evaluation , initial and 90-day modified Rankin Scale (mRS) assessments, Alberta Stroke Program Early CT Score (ASPECTS),and CT angiography (CTA)assessments before EVT.

GOOGLE SAYS

The 90-day modified Rankin Scale (mRS) is a widely used outcome measure in stroke clinical trials and clinical practice, assessing functional neurological disability after stroke at 90 days, ranging from 0 (no symptoms) to 6 (death). 

BUT THE TEXT IMPLIES THAT THE 90 DAY “MRS” WAS CONDUCTED BEFORE “EVT”.  DOES THAT MEAN THAT THE “EVT” WAS PERFORMED 90 DAYS AFTER THE STROKE?

MAYBE YOU MEAN

All patients had completed National Institutes of Health Stroke Score (NIHSS) evaluation , modified Rankin Scale (mRS) assessments, Alberta Stroke Program Early CT Score (ASPECTS),and CT angiography (CTA)assessments before EVT.  All patients were also evaluated with the 90-day modified Rankin Scale (mRS) and these other tests at 90 days after the EVT.

 Response 12: Thank you for pointing this out. We agree with this comment. Therefore, we have modified as instructed. The updated version appears in lines 157–166.

Comment 13:

Line 75

All patients had completed National Institutes of Health Stroke Score (NIHSS) evaluation , initial and 90-day modified Rankin Scale (mRS) assessments, Alberta Stroke Program Early CT Score (ASPECTS),and CT angiography (CTA)assessments before EVT.

WHENEVER AN EVALUATION METHOD IS INTRODUCED (LIKE NIHSS OR ASPECTS), PLEASE CITE THE BEST REFERENCE THAT DESCRIBES THE METHOD USED SO THAT A READER HAS ENOUGH INFORMATION TO DUPLICATE THE METHOD.

 Response 13: Thank you for pointing this out. We agree with this comment. Therefore, we have modified as instructed. The updated version appears in lines 157–158.

Comment 14:

Line 81

All patients underwent carotid sonography between the first to the second weeks after EVT

Line 96

To analyze the CBF in the treatment artery ….  blood flow

PERHAPS USING SONOGRAPHY THE VELOCITY WAS MEASURED IN THE DISTAL COMMON CAROTID ARTERY AND THE PROXIMAL INTERNAL CAROTID ARTERY IN THE NECK.  BUT IF THE “treatment artery” IS THE m2MCA, THEN THE MEASUREMENT IS DONE ON THE SIDE OF TREATMENT, NOT “in the treatment artery”.

 Response 14: Thank you for pointing this out. We could not check the FV of ACA or MCA.  We could only check the treatment side FV (ICA , CCA and ECA).

Comment 15:

TABLES

Variables in the tables are listed in an order.  The choice of the order is unclear, it is not based on R value (Pearson correlation), or on “p” values.  How was this order selected?

Response 15: Thank you for pointing this out. We agree with this comment. We corrected and listed our variables according to age, gender/ risk factors / before EVT parameter / t-PA /  during  EVT ..TICI/ post EVT ..sonography    

Comment 16:

Line 175

“selection for potentially good candidates for EVT”

THE PAPER DOES NOT DISCUSS HOW CASES WERE SELECTED FOR EVT.  IT LOOKS LIKE ONLY 10% OF CASES RECEIVED EVT AND OF THOSE, ABOUT HALF WERE INCLUDED IN THIS ANALYSIS. (Figure 1).

Response 16: Thank you for pointing this out. We agree with this comment. Therefore, we have modified as instructed. The updated version appears in lines 152–155.

Comment 17:

IT IS UNCLEAR FROM FIGURE 1 WHEN VARIOUS VARIABLES WERE MEASURED.

Response 17:

Thank you for pointing this out. The details have stated it accordingly in the section of study population section .The updated version appears in line 149-166.

Comment 18:

IN FIGURE 1, 172 PATIENTS WITH STROKE HAD EVT, 118 HAD OCCLUSION OF THE ICA OR MCA.

WHAT WAS THE SITUATION OF THE OTHER 54 PATIENTS?

HOW WAS THEIR SITUATION DISOVERED?

WHAT WAS THE PATHOLOGY OF THE MATERIAL RETRIEVED WITH EVT?

Response 18:  Thank you for pointing this out. The other 54 patients had occlusion of posterior circulation which was not the focus of our study. Thus, they were excluded. In all patients, CT perfusion was performed prior to EVT to evaluate vascular status, and digital subtraction angiography (DSA) was used intra-procedurally to confirm the occlusion site or assess vascular pathology. We did not routinely perform histopathologic analysis of the retrieved thrombi. That is a good point. Maybe we can analyze this in our future research.

Comment 19:

Line 175

carefully selection for potentially good candidates

I THINK THAT YOU MEAN

Carful selection of potentially good candidates

Response 19: Thank you for pointing this out. We agree with this comment. Therefore, we have modified as instructed. The updated version appears in lines 274-275.

Comment 20:

Line 182

we investigated multiple prognostic factors, including premorbid characteristics,

GOOGLE SAYS

Premorbid characteristics refer to an individual's functioning and personality traits before the onset of a disease or illness, particularly in the context of mental health conditions. 

I SUPPOSE THAT SEX (MALE), DIABETES AND SMOKING ARE PREMORBID AND THAT YOU DISCOVERED THESE IN THE MEDICAL CHART REVIEWS FOR DATES PRIOR TO THE STROKE EPISODE, BUT “r-tPA” is not a “premorbid condition”. 

Response 20: Thank you for pointing this out. We agree with this comment. Therefore, we have modified as instructed.

Comment 21:

Table 3

“Old stroke” MIGHT BETTER BE CALLED “Previous stroke”

Response 21:  Thank you for pointing this out. We agree with this comment. Therefore, we have modified as instructed. The updated version appears in table 1 and 2.

Comment 22:

ASPECTS and other diagnostic methods are not cited in the references.  This reviewer found

Stroke. 2017;48:1574-1579. DOI: 10.1161/STROKEAHA.117.016745 to learn about the head Computed Axial Tomography (also known as CT and CAT and EMI scan) method. 

Hmmm, GOOGLE is incorrect about EMI scan.  GOOGLE says

An EMI scan, which stands for ElectroMagnetic Interference, is a type of CT (Computed Tomography) scan developed by the company EMI, and it was the first to be adopted in substantial numbers for medicine, allowing for detailed pictures of the brain. 

BUT this is not correct.

the Beatles with their record company, EMI, indirectly led to the funding of the research that developed the CT scanner, also known as a CAT scan, by EMI engineer Godfrey Hounsfield

AND

Journal of Neurology, Neurosurgery, and Psychiatry, 1975, 38, 935-947

Mokin M, Primiani CT, Siddiqui AH, Turk AS, ASPECTS (Alberta Stroke Program Early CT Score) Measurement Using Hounsfield Unit Values When Selecting Patients for Stroke Thrombectomy,

(Stroke. 2017;48:1574-1579. DOI: 10.1161/STROKEAHA.117.016745.)

USES Hounsfield unit (HU) values on initial noncontrast head computerized tomography (CT) correlates with the extent of final infarct on follow-up imaging.

 Response 22:  Thank you for pointing this out. We have added another reference for the introduction of ASPECTS (Ref 16 )

Comment 23:

Line 214

we used the cerebral blood flow volume

Line 218

We regarded the flow ratio between the ICA and CCA as the flow required from the treated territory in the ipsilateral cerebral artery.

IN A STANDARD ULTRASONIC DOPPLER EXAMINATION OF THE CAROTID BIFURCATION, VELOCITIES (NOT FLOW RATES [VOLUME/TIME] OR VOLUME) ARE MEASURED (ESTIMATED USING THE DOPPLER EQUATION) FROM THE COMMON CAROTID ARTERY, INTERNAL CAROTID ARTERY AND EXTERNAL CAROTID ARTERY.  ALTHOUGH IT IS EXPECTED THAT THE FLOW THROUGH THE COMMON CAROTID ARTERY SHOULD BE EQUAL TO THE SUM OF THE FLOW THROUGH THE INTERNAL CAROTID ARTERY AND THE EXTERNAL CAROTID ARTERY, I DON’T THINK THAT THIS “CONSERVATION OF FLOW” HAS BEEN VERIFIED USING SONOGRAPHIC METHODS, OR ANY OTHER METHOD FOR THAT MATTER.  THE FLOW RATIO “TI/TC” MIGHT BE AN INDICATION OF FLOW DIVERSION THROUGH THE EXTERNAL CAROTID ARTERY AND ONTO THE BRAIN VIA THE OPHTHALMIC ARTERY.

 Response 23: :  Thank you for pointing this out.  If high degree stenosis of proximal  or distal ICA ( proximal MCA )happened ,reversed ophthalmic flow from the ECA may develop . The ratio of ICA/CCA FV may be indicative in such conditions.

Comment 24:

Good collateral status [5-8]

Line 198

After vessel recanalization, 3 conditions may develop. [12, 18, 19] The first condition is persistent stenosis or occlusion,

In healthy adults, TI/TC 222 ranged between 0.58 to 0.65. [26] [27]

  1. Scheel P, Ruge C, Schoning M. Flow velocity and flow volume measurements in the extracranial carotid and vertebral arteries in healthy adults: reference data and the effects of age. Ultrasound Med Biol. 2000;26(8):1261-1266. doi: 10.1016/s0301-5629(00)00293-3. PubMed PMID: 11120363.

  1. Schoning M, Walter J, Scheel P. Estimation of cerebral blood flow through color duplex sonography of the carotid and vertebral arteries in healthy adults. Stroke. 1994;25(1):17-22. doi: 10.1161/01.str.25.1.17. PubMed PMID: 8266366.

Flow velocity and flow volume measurements in the extracranial carotid and vertebral arteries in healthy adults: reference data and the effects of age.

Scheel P, Ruge C, Schöning M.

Ultrasound Med Biol. 2000 Oct;26(8):1261-6. doi: 10.1016/s0301-5629(00)00293-3.

PMID: 11120363

Color duplex measurement of cerebral blood flow volume in healthy adults.

Scheel P, Ruge C, Petruch UR, Schöning M.

Stroke. 2000 Jan;31(1):147-50. doi: 10.1161/01.str.31.1.147.

PMID: 10625730

  1. Schoning M,

4-8,9,10,11

CCA diametric pulsatility

TAV is time averaged velocity

Response 24: Thanks for these useful references  and so many recommendations. We have additional literature reviews on the clinical use of sonography and collateral flow in the discussion section.

Round 2

Reviewer 1 Report

Comments and Suggestions for Authors

This paper is a revised submission of a report on a single-centre retrospective cohort study of the association of extracranial carotid artery blood flow study performed in the second week after endovascular thrombectomy for acute ischaemic stroke, with functional outcome.

I thank the authors for revising, the paper is much improved.

There remain some issues that have not been fully addressed:

  1. Line 78 – please quote female rather than male (eg female 49%)
  2. Line 172 – use ‘second’ instead of ‘secondary’
  3. (major) Line 221 – as asked before, how do the 29 excluded subjects differ from the 89 included subjects? 25% is a high proportion of exclusions – it would be better to show a table comparing included and excluded patients, and p-values…
  4. Lines 304-313 – these can be absorbed into the Introduction
  5. (major) Discussion – in addition to THRIVE-EVT, as mentioned before, what about PRE, THRIVE-EVT, CLEAR, BET, BAND, PANDA scores?
  6. Line 450 – as mentioned before, additional limitations include the exclusion of 25% of patients…

Reviewer 2 Report

Comments and Suggestions for Authors

In Figure 1
Middle Cerebral Artery is usually abbreviated MCA
"anterior circulation artery (ACA)"
probably means the Anterior Cerebral Artery (ACA)
COMMENT
In usual terminology, the anterior cerebral circulation
includes the territory of the MCA and ACA.
That is the common interpretation of the term in the title
and on line 70 and 93 and 138 and 145 and 344 and 622.
On line 139, "anterior cerebral artery (ACA)" is used. 
On line 162 "anterior circulation artery" is deleted.

Line 291 Table 2  
QUESTION
In every case of "multivariate analysis", the "p" value
for the variable in multivariate analysis is larger than
in univariate analysis.  
Perhaps the multivariate analysis is intended to differentiate
essential variables from redundant variables.
Line 285  ASPECTS TICI (Thrombolysis in Cerebral Infarction) 
 treatment-side ICA CCA Blood Flow  TI/TC  
COMMENT
The reader might easily confuse abbreviations TICI with TI/TC.
In TI/TC, the "T" refers to time integrated flow
In other papers, flow is often called "Q".
In TICI, the "T" refers to thrombolysis.

Line 287 " remained independent predictors of fair outcomes"
VARIABLES
Continuous variables are "Age"  "TI/TC"
multiclass categorical variables are "NIHSS"  "ASPECTS"
[Mokin M, Primiani CT, Siddiqui AH, Turk AS. ASPECTS (Alberta Stroke Program Early CT Score) Measurement Using Hounsfield Unit Values When Selecting 
Patients for Stroke Thrombectomy. Stroke. 2017;48:1574-1579] Figure 2
yet Odds Ratios are presented for each.
The Alberta Stroke Program Early CT Score (ASPECTS) 
is a 10 category variable.
The National Institutes of Health Stroke Scale (NIHSS) 
is an 11 class, 2 to 4 option categorical summary variable
with a range from 0 to 42.
The statistical summary "mean+-SD" is based on the assumption
that the distribution of the values is ordinal and Gaussian.
"mean" is not an appropriate statistic for a categorical variable.

Since Table 2 depends primarily on Odds Ratio
Two changes might clarify the table:
1. eliminate the multivariate column and analysis in the paper
2. describe the conversion of all variables to binary
EXAMPLE Age<70yo; Age>70yo:: NIHSS<14.5; NIHSS>14.5
GOOGLE says QC=400 mL/min; QI=275 mL/min; QE=185 mL/min
Of course QI+QE=QC so these numbers don't add up.
But maybe a better parameter than the QI/QC="TI/TC" that you use
is the parameter QE/QI if directly measured.
From the GOOGLE numbers average QI/QC=275/400=0.69
Line 290 gives the threshold of 0.60.
Table 2 says means of 0.66 and 0.57.
Line 214 promises a "Receiver operating characteristic (ROC) curve analysis"
but that ROC figure is not in the manuscript. 

Line 306-308
"selection of potentially good candidates for EVT"
COMMENT
The selection of candidates is done before the EVT procedure
yet the manuscript promotes the use of Doppler ultrasound conducted 2 weeks after the EVT therapy.  It therefore cannot contribute to the 
EVT candidate selection.

COMMENT
If you have the "TI" (QI) and "TC" (QC) data, 
it would be interesting to see these on a figure (scattergram) of 
TI vs TC.  If you have the diameter and velocity data used for these parameters
it would be interesting to see the Diameter vs Time Integrated Velocity  scattergrams.
and also the Diameter vs Peak Velocity scattergrams.

Round 3

Reviewer 1 Report

Comments and Suggestions for Authors

This paper is a 2NDrevised submission of a report on a single-centre retrospective cohort study of the association of extracranial carotid artery blood flow study performed in the second week after endovascular thrombectomy for acute ischaemic stroke, with functional outcome.

I thank the authors for further revising, the paper is much improved.

There remains one issue that the authors could fully attend to – the excluded subjects. I suggest:

  1. Lines 425-428 ‘We excluded 29 patients who did not complete their sonography exam or follow-up. The comparison of baseline characteristics between these 29 patients and our enrolled cases can be found in supplementary Table 1.” can be cut and put in line 221 onwards, with comments where the 2 are statistically different (eg excluded subjects were less likely on NOACs, had higher NIHSS score, lower ASPECTS score, lower proportion of TICI 2b-3). The table can appear instead in the main text and not as a supplement
  2. Consequently, lines 426-428 ‘The comparison of baseline characteristics between these 29 patients and our enrolled cases can be found in supplementary Table 1.’ can be deleted, and a comment added on how this would impact on the Results

Author Response

Comment 1: Lines 425-428 ‘We excluded 29 patients who did not complete their sonography exam or follow-up. The comparison of baseline characteristics between these 29 patients and our enrolled cases can be found in supplementary Table 1.” can be cut and put in line 221 onwards, with comments where the 2 are statistically different (eg excluded subjects were less likely on NOACs, had higher NIHSS score, lower ASPECTS score, lower proportion of TICI 2b-3). The table can appear instead in the main text and not as a supplement

Response 1: Thank you for pointing this out. We agree with this comment. Therefore, we have modified as instructed. The updated version appears in lines 212-215 and 226-234.

Comment 2: Consequently, lines 426-428 ‘The comparison of baseline characteristics between these 29 patients and our enrolled cases can be found in supplementary Table 1.’ can be deleted, and a comment added on how this would impact on the Results

Response 2: Thank you for pointing this out. We agree with this comment. Therefore, we have modified as instructed. The updated version appears in lines 427-430.